# Proteome-Wide Analysis Reveals TFEB Targets for Establishment of a Prognostic Signature to Predict Clinical Outcomes of Colorectal Cancer

**DOI:** 10.3390/cancers15030744

**Published:** 2023-01-25

**Authors:** Zijia Huang, Sheng Zhu, Ziqin Han, Chen Li, Junze Liang, Yang Wang, Shuixing Zhang, Jing Zhang

**Affiliations:** 1Department of Radiology, The First Affiliated Hospital of Jinan University, Jinan University, Guangzhou 510613, China; 2MOE Key Laboratory of Tumor Molecular Biology, Key Laboratory of Functional Protein Research of Guangdong Higher Education Institutes, Institute of Life and Health Engineering, College of Life Science and Technology, Jinan University, Guangzhou 510632, China; 3Department of Nuclear Medicine, Affiliated Hospital of Xiangnan University, Xiangnan University, Chenzhou 423000, China

**Keywords:** TFEB, DIA-based proteomics, vesicular endocytic trafficking, colorectal cancer

## Abstract

**Simple Summary:**

Transcription factor EB (TFEB) is a master modulator of autophagy and lysosomal biogenesis. Dephosphorylation of TFEB at Ser142 and Ser138 determines its nuclear localization and transcriptional activity. The link between TFEB-associated genes and colorectal cancer (CRC) progression and prognosis remains unclear; thus, we performed data-independent acquisition (DIA)-based quantitative proteomics to systematically identify the targets of TFEB. Using stringent statistical criteria, 60 proteins associated with vesicular endocytic trafficking were identified as TFEB targets. Moreover, a prognosis-linked TFEB-related gene signature was developed, showing that patients with higher risk scores had higher epithelial–mesenchymal transition (EMT) scores with worse prognosis. Additionally, a nomogram was constructed by combining clinicopathological parameters and the gene signature to enhance the quantification capacity in risk assessment for individual patients. This research facilitates further mechanistic studies of TFEB, and the TFEB gene signature-based model may provide important information for assisting clinicians to predict CRC patient prognosis.

**Abstract:**

Dephosphorylation of transcription factor EB (TFEB) at Ser142 and Ser138 determines its nuclear localization and transcriptional activity. The link between TFEB-associated genes and colorectal cancer (CRC) progression and prognosis remains unclear. To systematically identify the targets of TFEB, we performed data-independent acquisition (DIA)-based quantitative proteomics to compare global protein changes in wild-type (WT) DLD1 cells and TFEB^WT^- or TFEB^S142A/S138A^ (activated status)-expressing DLD1 cells. A total of 6048 proteins were identified and quantified in three independent experiments. The differentially expressed proteins in TFEB^S142A/S138A^ versus TFEB^WT^ and TFEB^WT^ versus control groups were compared, and 60 proteins were identified as products of TFEB transcriptional regulation. These proteins were significantly associated with vesicular endocytic trafficking, the HIF-1 signaling pathway, and metabolic processes. Furthermore, we generated a TFEB-associated gene signature using a univariate and LASSO Cox regression model to screen robust prognostic markers. An eight-gene signature (*PLSCR3*, *SERPINA1*, *ATP6V1C2*, *TIMP1*, *SORT1*, *MAP2*, *KDM4B*, and *DDAH2*) was identified. According to the signature, patients were assigned to high-risk and low-risk groups. Higher risk scores meant worse overall survival and higher epithelial–mesenchymal transition (EMT) scores. Additionally, as per the clinicopathological parameters and gene signature, a nomogram was constructed that was utilized to enhance the quantification capacity in risk assessment for individual patients. This research shows that TFEB directly mediates network effects in CRC, and the identified TFEB gene signature-based model may provide important information for the clinical judgment of prognosis.

## 1. Introduction

Colorectal cancer (CRC), a common malignant tumor in the digestive system, ranks third in terms of incidence and second regarding mortality, according to global cancer statistics [1]. The molecular structure of CRC is extremely heterogenous and is therefore responsible for the heterogeneous and frequently suboptimal treatment response [2]. Thus, research has focused on molecular subtyping strategies based on single or multiomics data to categorize patients into subgroups to facilitate risk stratification and disease management [3].

Dysregulation of transcription factors (TFs) leads to pronounced changes in gene expression, which is a common phenomenon of human malignant neoplasias [4,5]. These alterations influence cell proliferation/differentiation, metastasis, and migration, as well as chemotherapeutic resistance, thereby acting as major factors concerning the behavior of tumors [6]. Transcription factor EB (TFEB), which belongs to the MiT-TFE helix TFs, performs a vital function in lysosomal biogenesis, autophagy, metabolism, and the immune response [7]. Numerous reports have indicated that TFEB is upregulated in various types of tumors [7,8], and high levels of TFEB expression in CRC tissues are associated with aggressive clinical characteristics and poor survival of CRC patients [9]. Overexpression of TFEB and upregulation of its transcriptional network appears to be sufficient to drive tumorigenesis in different tissues; therefore, modulation of the activity of TFEB can be considered as a potential therapeutic strategy [10].

The TFEB shuttling between the nucleus and cytoplasm is influenced by its phosphorylation [11]. After being translocated to the nucleus, TFEB activates its target gene’s transcription [12]. The nuclear export of TFEB is determined by the phosphorylation status of S142 and S138 near the nuclear export signal (NES). Dephosphorylation of S138/S142 when mTOR is inhibited results in TFEB nuclear retention [13]. The mutation of the aforementioned serines to alanines results in different TFEB types that appear to be localized in the nucleus and are constitutively active [12,13,14]. It is assumed that many reported targets of MiT members exert critical functions in tumorigenesis [7,15,16]; thus, identification of additional targets may be important for further insights regarding the functions of TFEB and its inhibition in cancer treatment. 

In this study, to systematically identify the targets of TFEB, we produced stably wild-type TFEB (TFEB^WT^)- or activated-status TFEB (TFEB^S142A/S138A^)-expressing CRC cells and quantitatively analyzed the proteomes of these cells. Using stringent statistical criteria, 60 proteins were identified as TFEB targets. Moreover, a prognosis-linked gene signature (TFEB related) was developed utilizing the robust biomarkers identified through the combination of different methods. These insights will help further identify TFEB targets and elucidate the functions of TFEB.

## 2. Materials and Methods

### 2.1. Cell Lines and Cell Culture

Human DLD1 and 293T cells were cultured as per the respective depositors’ recommendations after purchase from the American Type Culture Collection (Manassas, VA, USA).

### 2.2. Plasmid Construction, Retroviral Infection, and Transfection

TFEB^WT^-GFP and mutant TFEB^S142A/S138A^-GFP DNA sequences were amplified by PCR and were cloned into the pEGFP-N1 vectors, followed by subcloning into a pLVX-Puro lentiviral vector. Cells that overexpressed TFEB^WT^ or TFEB^S142A/S138A^ stably were generated as described previously [17]. Briefly, lipofectamine 3000 (Thermo Fisher Scientific, Waltham, MA, USA) was utilized to infect the cells by transfecting 293 T cells with virus skeleton vectors and lentiviral plasmids. CRC cells were infected using culture media containing viruses and were selected using puromycin. Prior to further experimentation, the selected stably overexpressed wild-type or mutant TFEB cells were maintained for at least two weeks.

### 2.3. Immunoblotting

Immunoblotting assays were performed as previously described [18]. SDS-PAGE was performed on the extracted proteins, which were then transferred onto PVDF membranes (BIO-RAD, Hercules, CA, USA). The gel bands formed after exposure to appropriate antibodies (primary and secondary) were visualized using the ECL reagent (BIO-RAD) and imaged by Tanon 5200-Multi (Tanon Science & Technology, Shanghai, China). The primary antibodies (against TFEB, GFP, LAMP2, NDRG1, SMPD1, and actin) and the HRP-conjugated secondary antibodies (goat anti-rabbit antibody) were supplied by Proteintech (Wuhan, China). Original uncropped images for immunoblotting are presented in Appendix A.

### 2.4. Cell Proliferation Assays

For the proliferation assay, CRC cells stably expressing TFEB^WT^ or TFEB^S142A/S138A^ were seeded in 96-well microplates at a density of 1.5 × 10^3^ cells per well and were cultured for four days. One day after seeding, cells were stained with CCK-8 (Beyotime, Jiangsu, China). The automated microplate spectrophotometer (BioTek Instruments, Winooski, VT, USA) was employed to read the plate after 1.5 h at the 450 nm wavelength. Absorbance was normalized against the absorbance on the first day and was calculated. Each experiment was performed in triplicate.

### 2.5. Confocal Examination

DLD1 cells stably expressing TFEB^WT^-GFP or TFEB^S142A/S138A^-GFP were plated on confocal dishes (NEST, Wuxi, China) and were incubated for 24 h. Prior to imaging, DAPI (Beyotime) was used as a stain to label the nucleus. TFEB^WT^ treated with Torin (TargetMol, Shanghai, China) was used as positive control. A confocal laser scanning microscope (LSM 880 with AiryScan, Carl Zeiss, Oberkochen, Germany) was utilized to visualize and record the GFP fluorescence.

### 2.6. Pinocytosis Assays

Pinocytosis was examined using a TRITC-dextran (70 kDa; Ruixi Biological Technology Co., Ltd., Xi’an, China) uptake assay. The incubation of the stably expressing TFEB^WT^-GFP or TFEB^S142A/S138A^-GFP DLD1 cells with 1 mg/mL TRITC-dextran was performed for 2 h at 37 °C, followed by thorough washing and processing for immunofluorescence assays as described above. The quantification of TRITC-dextran uptake was conducted through the “Analyze Particles” feature in Fiji software (Image J v1.4). Cells that absorbed TRITC-dextran were quantified with the aid of a flow cytometer (BD Biosciences, CA, USA) at 594 nm.

### 2.7. Mass Spectrometry (MS) Analysis

To assess global protein alteration, data-independent acquisition (DIA)-based proteomics was performed as per our prior study [19]. Briefly, cells were subjected to cell lysis buffer with subsequent trypsin digestion and desalination. Following the addition of iRT-Standard (Biognosys, Cambridge, MA, USA) to each sample, the Orbitrap Fusion Lumos mass spectrometer (Thermo Fisher Scientific) was employed for producing the DIA-MS raw data. The raw data were searched against the UniProt human protein database (http://www.uniprot.org, accessed on 20 January 2021). Among all mentioned alterations, the variable ones were considered to be the acetylation of lysine and the protein *N*-terminal as well as oxidation (M), whereas a fixed modification was considered to be carbamidomethylation (C). The peptide and protein identification false discovery rate cutoff values were set at 1%, and their expression levels were measured with the aid of the Spectronaut software v14 (Omicsolution, Shanghai, China). 

### 2.8. Analysis of Differentially Expressed TFEB Targets

The differentially expressed proteins (DEPs) in the TFEB^S142A/S138A^ and TFEB^WT^ groups (TFEB^S142A/S138A^/TFEB^WT^) and TFEB^WT^ and control groups (TFEB^WT^/control) were obtained using the R software v4.0.5 “limma” package [20], and the criteria used to define DEPs were as follows: fold change (FC) > 1.2 and *p* < 0.05. TFEB targets were identified as FC (TFEB^S142A/S138A^/TFEB^WT^) > FC (TFEB^WT^/control) > 1.2 and *p* < 0.05.

### 2.9. Gene Ontology Analysis

The enriched pathways of TFEB targets were identified through the R package “clusterProfiler” [21,22] utilizing GO (including CC: cellular component, BP: biological pathway, and MF: molecular function) and KEGG functional enrichment analyses. Utilizing the entire human genome as a reference, the functional annotation chart options set *p* < 0.05 for determining the significant biological pathways. 

### 2.10. Publicly Available mRNA Data

The relevant clinical information and RNA-seq data on colon cancer (COAD) from The Cancer Genome Atlas (TCGA) database were acquired by using the online tool Sangerbox (http://sangerbox.com/, accessed on 6 April 2022) [23], and were included in the training group. Subsequently, an expression matrix file (GSE17536) via the Gene Expression Omnibus (GEO) (http://www.ncbi.nlm.nih.gov/geo/, accessed on 20 May 2022) was utilized for external verification. 

### 2.11. Signature Generation

The “Survival” R package was employed for univariate Cox regression analysis to screen out the differentially expressed TFEB targets linked to prognosis, whereas further operations such as the Cox regression model of the least absolute shrinkage and selection operator (LASSO) were fit using the “Glmnet” R package to further identify the most robust candidates. The gene expression levels were normalized and calculated by the corresponding LASSO Cox coefficients, which were then used to establish the TFEB-related risk score (TRS) as follows: TRS = ∑_i_ Coefficient (mRNA_i_) × Expression (mRNA_i_)

### 2.12. Classification, Prediction, and Validation in GEO and TCGA

As per the formula of the signature, the patients were classified into two risk groups (high-risk and low-risk groups) depending on the TRS values. The threshold values and area under the curve were obtained and analyzed using the receiver operating characteristic (ROC), a time-dependent function. The aforementioned values were calculated for 1-, 3-, and 5-year overall survival (OS) and relapse-free survival to validate the performance of the signature using the “survivalROC” R package. The predictive significance of the risk score formula concerning prognosis was assessed using log-rank tests and Kaplan–Meier (K-M) survival curve analyses. Time-dependent concordance index (C-index) and time-dependent receiver operating characteristic (tROC) analyses were used to compare the predictive capacity of survival among different variables using the R packages “survConcordance” and “survivalROC”.

### 2.13. Gene Set Enrichment Analysis (GSEA)

GSEA Software V3.0 (http://software.broadinstitute.org/gsea/index.jsp, accessed on 18 August 2022) [24] and h.all.v7.4.symbols.gmt gene set (http://software.broadinstitute.org/gsea/index.jsp, accessed on 18 August 2022) [24] were used to investigate the associated pathways in different risk subgroups. Based on the gene expression profile in TCGA and phenotypic grouping, the minimum gene set was set as 5, and the maximum gene was set as 5000. After a thousand resamples, the functional annotation chart options set *p* < 0.05 for determining the significant hallmark gene sets. 

### 2.14. Calculation of Epithelial–Mesenchymal Transition (EMT) Scores

The EMT scores were calculated as previously reported by Milena P. Mak et al. [25], using the pan-cancer EMT signature (Appendix A [25]). For each CRC patient, the EMT score was calculated as the mean expression of the mesenchymal markers minus the mean expression of the epithelial markers.

## 3. Results

### 3.1. Constitutive Nuclear Import of TFEB Promotes CRC Proliferation

Dephosphorylation of the two highly evolutionarily conserved serines S138 and S142, which are situated near a NES (Figure 1A,B), is critical for its nuclear retention and activation. To examine the role of activated TFEB in CRC, a series of constructs was generated in which the GFP tag was fused to TFEB^WT^ or TFEB^S138A/S142A^. Substantial expression of the TFEB-GFP fusion protein was observed in TFEB^WT^- and TFEB^S138A/S142A^-transfected cells through Western blotting using anti-TFEB and anti-GFP antibodies (Figure 1C). Confocal microscopy showed that mutations in S138 and S142 caused almost complete nuclear retention of TFEB (Figure 1D,E), as compared to TFEB^WT^, similar to the mTOR inhibitor Torin. The CCK-8 assay showed that CRC cells overexpressing both WT and mutant TFEB displayed a pro-proliferation phenotype, as compared to the control, while TFEB^S138A/S142A^ displayed a higher proliferation ability than TFEB^WT^ cells (Figure 1F), suggesting an oncogenic role of activated TFEB in CRC.

### 3.2. DIA-Based Proteomics Profiling Reveals Potential TFEB Targets

To systematically identify downstream proteins regulated by TFEB, the proteome of DLD1-TFEB^WT^, -TFEB^S142A/S138A^, and control cell lines were analyzed through DIA-based proteomics (Figure 2A). Proteins from these lines were profiled in triplicate, and 6048 proteins (Appendix A) were identified and quantified at 1% FDR. Ridge plots were employed to show the expression distribution of these proteins in different groups (Figure 2B), and no significant difference was observed among each set of triplicates; however, the expression distribution differed between the three groups (*p* = 9.7 × 10^−4^). To examine the difference between the proteins measured in the three groups and to validate the reproducibility of the triplicate measurements in each experiment, a PCA was performed. This revealed that the samples could be divided into TFEB^WT^, TFEB^S142A/S138A^, and control groups, thus confirming the similarity of each set of triplicate measurements (Figure 2C). We analyzed DEPs in TFEB^S142A/S138A^ versus TFEB^WT^ and TFEB^WT^ versus control groups through volcano plots and obtained a list of 480 upregulated and 964 downregulated proteins in the TFEB^S142A/S138A^/TFEB^WT^ groups and 982 and 117 in the TFEB^WT^/control groups (>1.2-fold change, *p* < 0.05; Figure 2D, Appendix A). To better identify and recognize targets transcriptionally regulated by TFEB translocation to the nucleus without being affected by TFEB overexpression, we selected proteins that were upregulated in the TFEB^WT^/control groups and remained upregulated when the proteome of TFEB^S142A/S138A^ cells was compared with that of TFEB^WT^ cells (Figure 2E). After filtering, 60 proteins showed gradually increased expression levels, in the order of control, TFEB^WT^, and TFEB^S142A/S138A^ (Figure 2F, Appendix A). Subsequently, three randomly selected proteins (LAMP2, NDRG1, SMPD1) from the 60 proteins were verified by immunoblotting and qRT-PCR. The results showing the quantitative trend of the three DEPs in MS were consistent with the immunoblotting and qRT-PCR results (Appendix A). The high accuracy of our MS result provided a basis for the subsequent analysis of the 60 TFEB-regulated proteins. 

### 3.3. TFEB Increases Cellular Pinocytosis Rates

To further characterize the 60 TFEB targets, KEGG and GO analyses were performed. The KEGG analysis showed that the TFEB targets significantly regulated lysosome-related cellular processes, the HIF-1 signaling pathway, and metabolism (Figure 3A). According to the GO analysis, these TFEB targets were enriched in metabolic and biosynthetic processes under biological progress (Figure 3B). The endoplasmic reticulum, granule, vesicle, vacuole, and early endosome were the main cellular components (Figure 3C). Oxidoreductase activity made up a high proportion of the molecular function (Figure 3D).

The CC enrichment analysis results of this research were consistent with previous reports, where TFEB-mediated endocytosis and phagocytosis could promote tumor growth. Pinocytosis is similar to phagocytosis and endocytosis in terms of being a lysosome-dependent and evolutionarily conserved pathway and offers an alternative nutrition acquisition pathway via which malignant cells consume (take up) extracellular substances. Science pinocytosis is not included in GO terms; to test whether TFEB may also affect pinocytosis, we determined pinocytosis rates by measuring the TRITC-labeled dextran uptake. Confocal microscopy showed that both TFEB^WT^ and TFEB^S138A/S142A^ cells had enhanced extracellular TRITC-dextran uptake as compared to the control, while TFEBS^138A/S142A^ displayed a higher TRITC-dextran uptake ability than TFEB^WT^ cells (Figure 3E). The effect of TFEB on pinocytosis was confirmed by flow cytometry and revealed that the TRITC-dextran uptake ability of the TFEB^WT^ and TFEB^S138A/S142A^ cells was consistent with the results obtained by confocal microscopy (Figure 3F). Collectively, these results suggested that TFEB is involved in the regulation of all three general modes of vesicular endocytic trafficking including pinocytosis.

### 3.4. Establishment of a TFEB Target Signature for Prognosis

The 60 TFEB targets were analyzed through univariate Cox regression, and promising candidates with a cutoff value of the log-rank test of *p* < 0.05 were utilized for robust prognostic marker detection by inclusion in the LASSO Cox regression model. Overfitting was eliminated by tenfold cross-validation, with an optimal λ value of 0.007 (Figure 4A). A set of eight genes (*PLSCR3*, *SERPINA1*, *ATP6V1C2*, *TIMP1*, *SORT1*, *MAP2*, *KDM4B*, and *DDAH2*) as well as their individual LASSO coefficients were generated (Figure 4B). Their corresponding hazard ratios and LASSO coefficient distributions are listed in Figure 4C,D. As a result, the TRS formula: TRS = ∑_i_ Coefficient (mRNA_i_) × Expression (mRNA_i_) was generated.

We next divided the CRC patients from the training set into high- and low-risk groups according to their TRS. The expression levels of risk-type genes (*PLSCR3*, *ATP6V1C2*, *TIMP1*, *MAP2*, *KDM4B*, and *DDAH2*) were elevated in the high-risk category in contrast with the low-risk category, whereas the expression levels of protective-type genes (*SERPINA1* and *SORT1*) had the opposite trend (Figure 4E). Ranked by TRS, the CRC patients were examined regarding the link between survival times and survival status (Figure 4F). The eight genes’ expression patterns were represented by a heatmap, which demonstrated the predictive ability of the eight-gene signature in the prognostic values in patients as well as their potential influence on the onset and progression of malignancies.

A higher TRS was associated with a poorer prognosis for patients than lower scores, according to a K-M analysis (HR = 2.91, *p* = 1.2 × 10^−7^; Figure 4G). The prognostic model’s predictive ability was tested using the ROC curves; the AUCs of these curves were estimated at around 0.7, indicating good predictive performance (Figure 4H). The same model was utilized to verify the eight-gene signature in the testing dataset (GSE17536, *n* = 177). The K–M survival curves depicted increased survival rates for the low-risk category (Figure 4I), which was congruent with the training group data. The ROC curves depicted diverse AUC values ranging from 0.61 to 0.7 (Figure 4J). Collectively, these results suggested that the models based on the eight-gene signature had high specificity and sensitivity. 

### 3.5. Tumor-Biology-Associated Potential Functions of the TFEB Signature

To gain further insights into the features of the TFEB signature, the link between the cancer hallmark gene sets and TRS was examined (Figure 5A). The group with high risk showed enrichment of the EMT, angiogenesis, myogenesis, hedgehog signaling, apical junction, notch signaling, and coagulation as per the GSEA data, whereas protein secretion was reduced. Among these enriched pathways, EMT produced the highest enrichment score; therefore, EMT scores for the CRC patients were subsequently calculated. Compared to normal tissues, cancer tissues had increased EMT scores (Figure 5B), and the high-risk group had higher EMT scores (Figure 5C). Furthermore, the EMT score positively correlated with the TRS (Pearson’s *R* = 0.41, *p* = 4.3 × 10^−19^, Figure 5D). These results indicated that EMT made pronounced contributions to the prognosis of COAD patients with a higher TRS.

### 3.6. Combination of the TRS and Clinicopathological Characteristics Improves Survival Prediction

To compare the TRS with the crucial clinical factors, we analyzed the TRS differences between subgroups with distinct clinicopathological features. In the TCGA dataset, T3–4, stage III–IV, and N1–2 patients had an increased TRS (Figure 6A); similarly, in the testing dataset GSE17536, the TRS was increased in stage 3–4 and grade 3 (Appendix A). To enhance the quantification capacity in risk assessment, we constructed a nomogram using the TRS and six clinical features including gender; age; clinical N, M, and T; and stage (Figure 6B). The actual 3-year survival times were congruent with the anticipated survival times, as per the calibration plots (Figure 6C). The K–M survival analyses indicated significant OS in the risky group (Figure 6D). The prediction capacity of the nomogram was assessed through tROC analyses, with an average AUC above 0.8, which was markedly better than pathological TNM staging. (Figure 6E). Furthermore, the nomogram also showed a robust performance in the prediction of survival rates in the testing dataset (Appendix A). These results suggested that the TRS-based nomogram had a stable and powerful performance in the prediction of survival rates.

## 4. Discussion

Aberrant regulation of MiT/TEF transcription factors plays a critical role in tumorigenesis [7,8,9,10,12]; however, the network directly mediated by TFEB in CRC remains obscure. In the current study, we produced TFEB^S138A/S142A^, a continuously activated form, to systematically investigate direct targets of TFEB by performing in-depth DIA-MS-based quantitative proteomics. A total of 60 TFEB-regulated DEPs were identified, and a univariate and LASSO Cox regression model was performed to select robust prognostic biomarkers for the creation of a TFEB-related gene signature. Additionally, the accuracy and prediction capability were enhanced through the construction of a nomogram model according to the clinicopathological features and TFEB-related gene signatures.

As a master transcriptional regulator of cell catabolism, TFEB is translocated from the cytoplasm to the nucleus to control gene expression that is relevant to lysosomal and autophagosomal biogenesis [12,26]. Our results consistently showed that TFEB carried Ser-to-Ala mutations of both Ser138 and Ser142 and showed constitutive nuclear localization, which significantly promotes CRC growth. The activation of TFEB influences various downstream cellular processes that are vital for phagocyte functions, such as lysosome functioning and autophagy, endocytosis, phagocytosis, endoplasmic reticulum restoration, mitochondrial homeostasis, lipid and glucose metabolism, and the production of cytokine and chemokine [27]. In congruence with these results, we found that these proteins were localized in the endoplasmic reticulum and structures associated with secretion and endocytosis, and they were involved in the regulation of lysosome processes, metabolic processes, HIF-1 signaling, oxidoreductase activity, and several binding functions, indicating that the functions of TFEB are not limited to autophagy and lysosome biogenesis.

Vesicular endocytic trafficking is essential in regulating tumor metastasis and growth. As per the type of cargo, internalization route, and scission mechanism, vesicular endocytic trafficking occurs in three general types in the cell: pinocytosis, phagocytosis, and endocytosis [28]. TFEB induces cellular endocytosis and phagocytosis to mediate lysosomal biogenesis, autophagy flux, and MTORC1 signaling [27,29]. In this research, the cellular components of TFEB targets were linked to vesicle membranes during phagocytosis and endocytosis, and an unidentified TFEB function of enhancing the pinocytosis pathway was detected (Figure 3E,F), indicating that TFEB is involved in regulating all three general modes of vesicular endocytic trafficking. These findings may help explain why active TFEB can promote CRC cell proliferation (Figure 1E) and pronounced TRS group enrichment in the carcinogenic activation pathways (Figure 5).

Based on the identified TFEB-associated genes, we established a prognostic CRC signature with a combination of eight genes (*PLSCR3*, *SERPINA1*, *ATP6V1C2*, *TIMP1*, *SORT1*, *MAP2*, *KDM4B*, and *DDAH2*) and an equation for calculating a TRS (Figure 4). The poorer prognosis observed in the group with higher TRS values through survival analysis was confirmed using independent cohorts. Additionally, GSEA analysis revealed considerably increased enrichment levels concerning the carcinogenic activation pathways in the group with higher TRS values, i.e., EMT, angiogenesis, myogenesis, hedgehog signaling, apical junction, notch signaling, and coagulation. These findings provided a more thorough and credible reason for the worse prognosis in patients belonging to the high-TRS subgroup. Additionally, a nomogram was generated containing the TRS and other clinicopathological parameters. In comparison with various characteristics, a stable and powerful performance was shown by the nomogram regarding the prediction of survival rates during follow-up at different times (Figure 6). The resulting data indicated the high potential of the TRS for CRC patient prognosis, and the capability of the nomogram as a reliable method for diagnosing CRC patients in clinical settings. 

Six of the eight genes served as risky types in our study, and most of them have been studied in many cancers. For example, involved in both tumor progression and metastasis, *ATP6V1C2* functions in the biological process of transferring hydrogen ions [30]. *TIMP1*, termed tissue inhibitor matrix metalloproteinase 1, has been widely studied in various cancers. In colorectal cancer, *TIMP1* promotes metastasis through the FAK-PI3K/AKT and MAPK pathways [31]. *MAP2*, a crucial regulator of microtubule dynamics, exerts dual effects on cancer progression [32]. Our research revealed that *MAP2* correlated with a poor prognosis of CRC and served as a risk biomarker. *KDM4B* promotes the progression of colorectal cancer and glucose metabolism by promoting TRAF6-mediated AKT activation [33]. *DDAH2* was reported to be involved in the invasion of lung adenocarcinoma by promoting angiogenesis [34]. The expression of *SERPINA1* and *SORT1* has previously been shown to be increased in several types of cancers [35,36]; however, the two proteins served as two protective biomarkers in the current study. Notably, *PLSCR3*, with the highest risk coefficient in our model, has rarely been investigated in tumor research. Therefore, more research on CRC is needed to fully understand the molecular processes related to the TFEB of the novel gene signature. 

## 5. Conclusions

Taken together, we performed a proteome-wide analysis of potential TFEB targets, and our insights facilitate further mechanistic studies of TFEB. Furthermore, a novel TFEB-regulated gene signature was established to identify CRC patients with increased risk. By incorporating this with clinicopathological characteristics, we assembled a nomogram to quantitatively assess individual patients’ risk scores. The TFEB gene signature-based model may prove to be efficient in assisting the clinician to predict patient prognosis, thereby leading to the effective management of CRC on an individual basis.

## Figures and Tables

**Figure 1 cancers-15-00744-f001:**
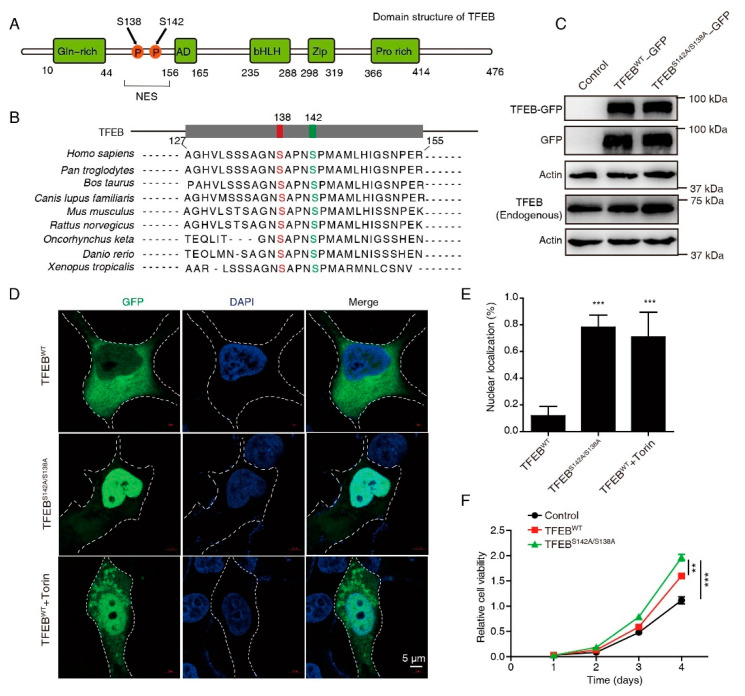
Mutations in S138 and S142 caused almost complete nuclear retention of TFEB. (**A**) Schematic of the human TFEB protein domains. S138 and S142 are localized in the proximity of a NES. (**B**) S138 and S142 of TFEB are evolutionarily conserved in the indicated species. Alignment of the sequences near TFEB S138 and S142 is shown. (**C**) Western blots of endogenous and exogenous TFEB in CRC cells transfected with wild-type TFEB-GFP or TFEB-GFP mutant (S142A/S138A). Uncropped immunoblots are provided in the Appendix A. (**D**,**E**) Localization of TFEB^WT^ and TFEB^S142A/S138A^ in the cytosol and in the nucleus was detected by confocal microscopy. Representative images (**D**) and statistical results (**E**) are shown. Torin1 (250 nm), a positive control. Cell outlines in white dotted lines. DAPI was used to label the nucleus. Scale bars, 5 μm. Mean ± SD, *n* = 10 cells per condition, unpaired *t*-test. (**F**) Cell viability of indicated cells were analyzed using a CCK8 assay. Mean ± SEM, *n* = 3, unpaired *t*-test. ** *p*  <  0.01, *** *p*  <  0.001.

**Figure 2 cancers-15-00744-f002:**
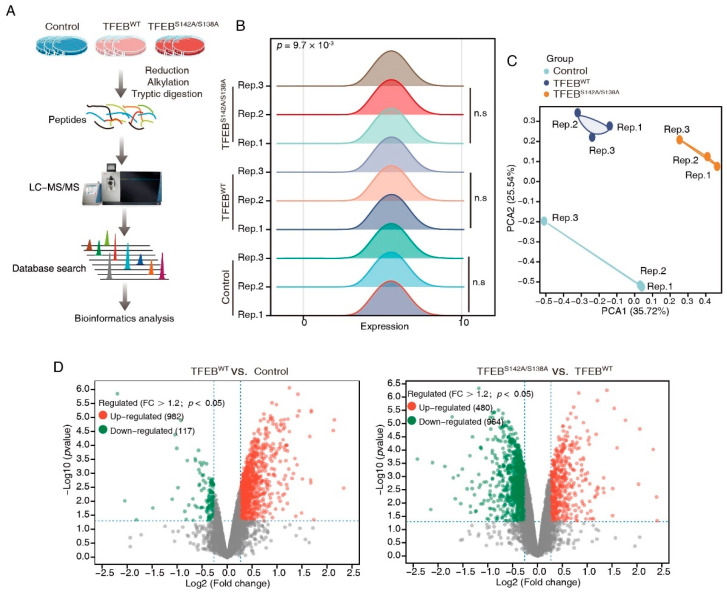
DIA-based quantitative proteomics characterizes TFEB targets. (**A**) The workflow of three independent DIA-MS experiments for identification of the TEFB-regulated proteins. (**B**) Ridge plots showing protein expression distribution in each group. (**C**) PCA analysis demonstrating that samples in the control, TFEB^WT^, and TFEB^S142A/S138A^ were separated into three discrete groups. (**D**) Volcano plot showing the DEPs in TFEB^S142A/S138A^ versus TFEB^WT^ and TFEB^WT^ versus control comparisons. (**E**) Venn diagram showing 60 TFEB targets. (**F**) Heatmap showing the expression of TFEB-regulated proteins in the control, TFEB^WT^, and TFEB^S142A/S138A^, groups.

**Figure 3 cancers-15-00744-f003:**
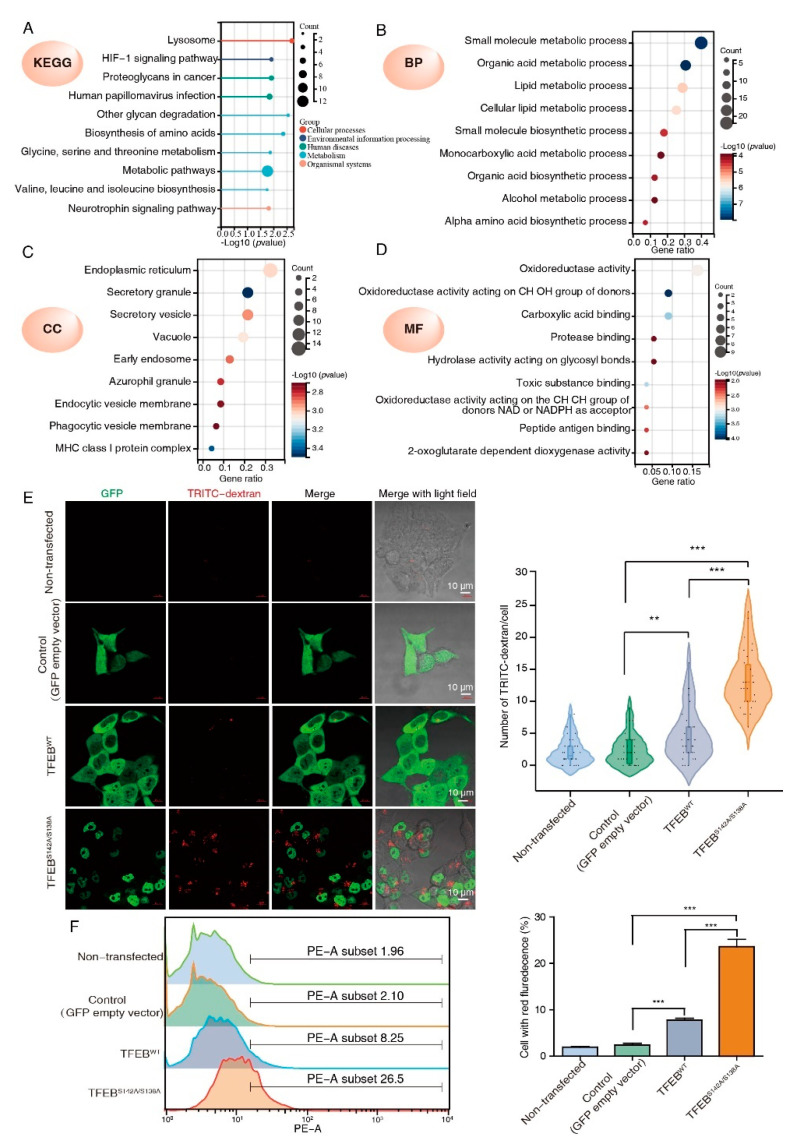
GO classification and pathway analysis of the TFEB transcriptional regulated proteins. (**A**) KEGG analysis of the 60 TFEB transcriptional regulated proteins. (**B**–**D**) The same proteins were subjected to the GO analysis (BP, biological process; CC, cellular component; MF, molecular function). (**E**,**F**) DLD1 cells stably expressing TFEB^WT^ or TFEB^S142A/S138A^ were incubated with TRITC-dextran (0.2 mg/mL) in a growth medium at 37 °C for 2 h. The representative confocal images show internalized TRITC-dextran (red dots) in indicated cells. Scale bars, 10 μm. Quantification is indicated on the right. Mean ± SD, *n* = 30 cells per condition, unpaired *t*-test. (**E**) Pinocytosis quantification using TRITC-dextran in indicated cells by flow cytometry. (**F**) Mean ± SEM, *n* = 3, unpaired *t*-test. ** *p*  <  0.001, *** *p*  <  0.001.

**Figure 4 cancers-15-00744-f004:**
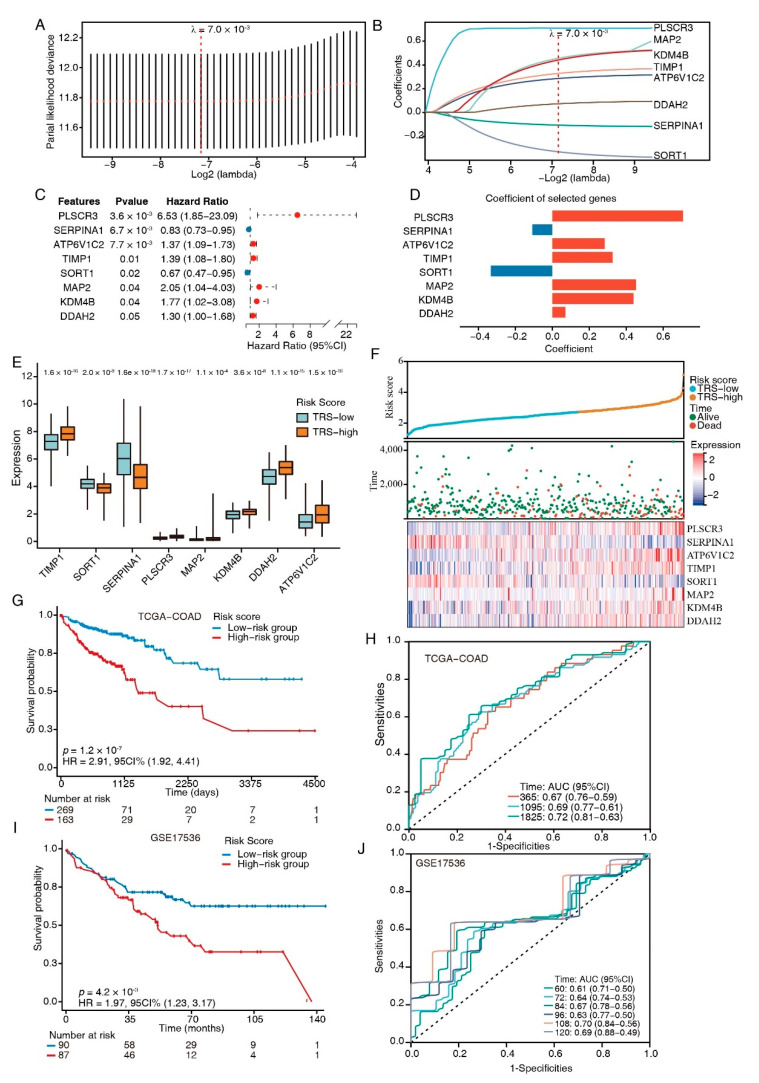
TFEB-related gene signature establishment. (**A**) Selection of the optimal LASSO parameter lambda; vertical lines indicate the optimal values. (**B**) LASSO coefficient profiles of the 8 candidate genes. (**C**) Forest plots of the 8-gene signature. (**D**) Coefficients of the eight genes. (**E**) Boxplots showing the 8-gene signature expression levels between high- and low-risk-score groups. (**F**) The distributions of the risk score and survival status of CRC patients, and the heatmap showing the eight genes’ expression profiles. (**G**) K–M survival curves for patients at high and low risk in the TCGA-COAD cohort. (**H**) ROC plot and AUC scores for predicted OS by the TRS in the TCGA-COAD cohort. (**I**) Kaplan–Meier survival curves of OS between high- and low-risk patients in the GSE17536 dataset. (**J**) ROC plot and AUC scores for predicted OS by the TRS in the GSE17536 dataset.

**Figure 5 cancers-15-00744-f005:**
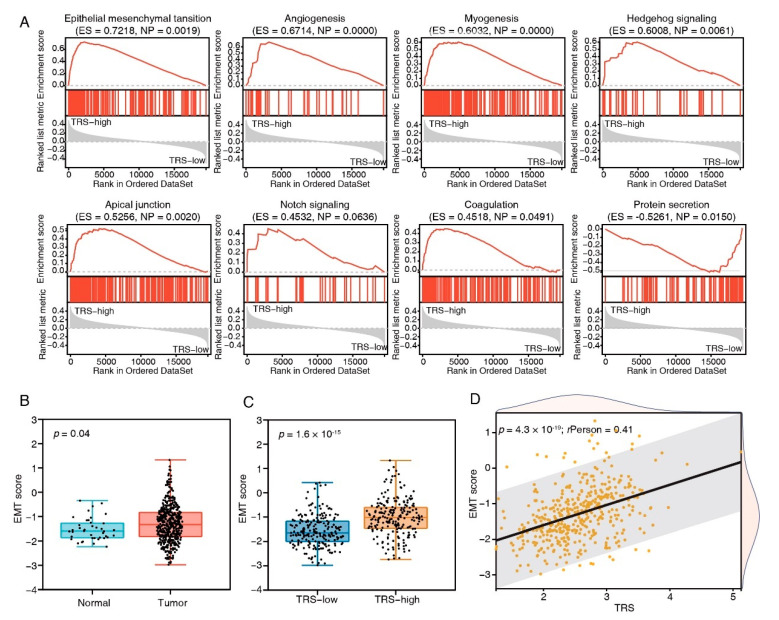
Biological pathways of TFEB score subtypes. (**A**) GSEA enrichment plots showing the 8 most enriched cancer hallmark pathways *(p <* 0.05). (**B**) The comparison of EMT scores in tumor and normal tissues in the TCGA-COAD cohort. (**C**) The comparison of EMT scores in high- and low-risk groups in the TCGA-COAD cohort. (**D**) The correlation analysis between EMT scores and risk scores.

**Figure 6 cancers-15-00744-f006:**
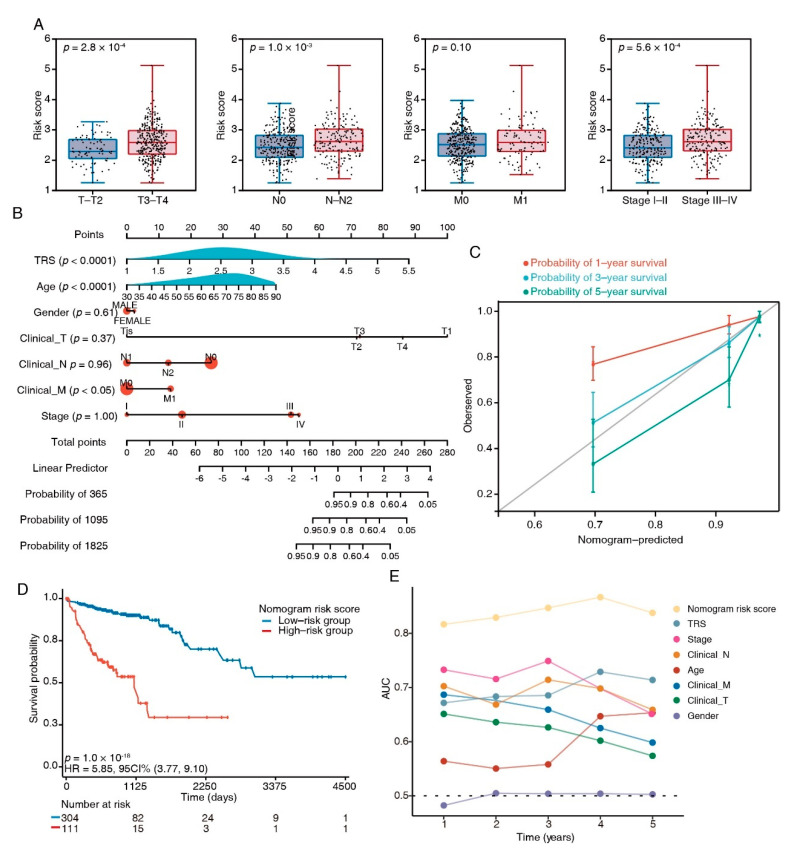
Nomograms showing the results of prognostic models using the TFEB signature and TNM staging system to predict OS of patients with CRC. (**A**) Distribution of TRS in clinicopathological variables, e.g., tumor (T), lymph node status (N), metastasis (M), and clinical stage. (**B**) Nomogram based on risk score and clinical factors. (**C**) Calibration plots of the nomogram for predicting the probability of 1-, 3-, and 5-year survival, ● represents the mean and * represents the media. (**D**) Kaplan–Meier curves of OS between high-risk and low-risk patients in the TCGA-COAD cohort. (**E**) tROC analysis to assess the accuracy of the nomogram.

## Data Availability

All data are available from the corresponding authors upon request.

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
