# Peer review of "Proteome-Wide Analysis Reveals TFEB Targets for Establishment of a Prognostic Signature to Predict Clinical Outcomes of Colorectal Cancer"

_cancers, 2023, doi:10.3390/cancers15030744_

Round 1

Reviewer 1 Report

Commments to the authors

This manuscript by Huang Z and coworkers focuses the reader’s attention on the TFEB directly mediating network effects during CRC by performing DIA-based quantitative proteomics and bioinformatic analysis. The authors performed DIA-MS on stably TFEBWT- or activated status TFEBS142A/S138A-expressing CRC cells to characterize potential TFEB targets and found that these TFEB-related proteins were involved in regulating all three general modes of vesicular endocytic trafficking including pinocytosis, phagocytosis, and endocytosis. Furthermore, the authors established a novel TFEB-regulated gene signature to identify CRC patients with increased risk. By incorporating this with clinicopathological characteristics, they assembled a nomogram to enhance the quantification capacity in risk assessment for individual CRC patients. This research facilitates further mechanistic studies of TFEB, and the TFEB gene signature-based model may provide critical information for assisting the clinician in predicting CRC patient prognosis. I am generally satisfied with the methods used, the described results, the summarized conclusion and the way and the story is presented. I only have relatively minor comments. See below:

Minor points:

- The method for calculating EMT scores needs to be described in the Materials and methods.

- Figure 1D-E, Figure 3E-F, please indicate the statistical method in the figure legends.

- Figure 4E-H seems to have with wrong figure legends order. Please corrected.

- Please provide more updated references for the targets of MiT members exerting critical functions in tumorigenesis in the introduction section (i.e., PMID: 34936535, 35970822……).

Reviewer 2 Report

In the manuscript entitled “Proteome-Wide Analysis Reveals TFEB Targets for Establishment of a Prognostic Signature to Predict Clinical Outcomes of Colorectal Cancer” , Zhang and colleagues reported that TFEB directly mediates network effects during CRC, and the identified TFEB gene signature-based model may provide important information for the clinical judgment of prognosis.

Overall the work reported potentially interesting.

Comments:

1)   The level of expression of TFEB in CRC is lower than normal tissue in both mRNA and protein level doi:10.2147/OTT.S180112. What is the rational of the study?

2)   Fig1C, please provide endogenous western data

3)   Fig1D,

ü  no DAPI staining;

ü  The author described Torin as a positive control. Please explain;

ü  How did the graph showing nuclear localization being generated?

4)   Fig2, please provide validation of the data, mRNAs and protein level.

5)   Fig3EF, please add Control, i.e., no transfected cells.

6)   Fig6, please validate the data using in house cohort.

Round 2

Reviewer 2 Report

The ms is much improved. no more comments.